# Bacterial coinfection restrains antiviral CD8 T-cell response via LPS-induced inhibitory NK cells

Tobias Straub[1], Marina A. Freudenberg[2,3], Ulrike Schleicher[4,5], Christian Bogdan[4,5], Georg Gasteiger[6,7] & Hanspeter Pircher[1]

Infection of specific pathogen-free mice with lymphocytic choriomeningitis virus (LCMV) is a widely used model to study antiviral T-cell immunity. Infections in the real world, however, are often accompanied by coinfections with unrelated pathogens. Here we show that in mice, systemic coinfection with *E. coli* suppresses the LCMV-specific cytotoxic T-lymphocyte (CTL) response and virus elimination in a NK cell- and TLR2/4-dependent manner. Soluble TLR4 ligand LPS also induces NK cell-mediated negative CTL regulation during LCMV infection. NK cells in LPS-treated mice suppress clonal expansion of LCMV-specific CTLs by a NKG2D- or NCR1-independent but perforin-dependent mechanism. These results suggest a TLR4-mediated immunoregulatory role of NK cells during viral-bacterial coinfections.

[1] Institute for Immunology, Medical Center – University of Freiburg, Faculty of Medicine, University of Freiburg, 79104 Freiburg, Germany. [2] Institute for Biology III, Faculty of Biology, University of Freiburg, 79104 Freiburg, Germany. [3] Department of Pneumology, Medical Center – University of Freiburg, Faculty of Medicine, University of Freiburg, 79104 Freiburg, Germany. [4] Mikrobiologisches Institut - Klinische Mikrobiologie, Immunologie und Hygiene, Universitätsklinikum Erlangen, Friedrich Alexander-Universität (FAU) Erlangen-Nürnberg, 91054 Erlangen, Germany. [5] Medical Immunology Campus Erlangen, FAU Erlangen-Nürnberg, 91054 Erlangen, Germany. [6] Institute of Systems Immunology, University of Wuerzburg, 97078 Wuerzburg, Germany. [7] Institute for Medical Microbiology and Hygiene, University of Freiburg Medical Center, 79104 Freiburg, Germany. Correspondence and requests for materials should be addressed to H.P. (email: hanspeter.pircher@uniklinik-freiburg.de)

nfection of mice with lymphocytic choriomeningitis virus (LCMV) is widely used to study innate and adaptive immune responses. Depending on viral strain and virus dose, LCMV induces either an acute infection, characterized by a potent virus-specific cytotoxic CD8+ T-lymphocyte (CTL) response followed by rapid virus clearance, or a chronic infection with T cell exhaustion and virus persistence. In most cases, clean laboratory mice kept under specific pathogen-free (SPF) conditions have been used for these studies. However, viral infections in real life may be accompanied by coinfections with unrelated pathogens that have the potential to modulate anti-viral immune responses[1]. The impact of a LCMV infection on a coinfection with bacterial pathogens has been analyzed in a number of studies. These data show that the LCMV infection can aggravate secondary infections with certain bacteria but may also protect against Gram-positive pathogens[2–4]. Enhanced susceptibility of LCMV-infected mice to LPS treatment has also been reported[5–7]. However, the reverse scenario, i.e., the effect of a bacterial coinfection on LCMV-specific T-cell immunity, has so far only been analyzed in a polymicrobial sepsis model[8]. These experiments showed that sepsis induced by cecal-ligation and puncture strongly impaired subsequent induction of a LCMV-specific CTL response[9–12]. Mechanistically, these findings have been explained by apoptosis-induced loss of antigen presenting cells[12], decrease in LCMV-specific precursor T-cells[10], alterations in memory CD8 T-cell function[11] or exacerbation of T-cell exhaustion[9].

NK cells are well-known for their potent antiviral and anti-tumoral activity but it is also evident that they function as important regulators of adaptive immunity during viral infections. In the murine cytomegalovirus (MCMV) infection model, NK cell-depletion prior to infection has been shown to improve T-cell responses and consequently virus elimination[13–15]. For infection with LCMV, which is not primarily controlled by NK cells, it was demonstrated that NK cells suppress antiviral immunity by killing activated CD4 and CD8 T-cells[16–18]. Accordingly, ablation of NK cells before or during chronic LCMV infection led to a stronger T-cell response and more efficient virus clearance[19,20]. By suppressing the CD4 T-cell response, NK cell regulatory activity also effects immune memory and B cell immunity during LCMV infection[21,22]. Importantly, these regulatory activities of NK cells during LCMV infection were only observed when high (>10^4 pfu) infectious doses were used for inoculation. In low dose (200 pfu) infection settings, NK cell depletion did not improve the LCMV-specific CTL response and virus clearance[23–25].

NK cells activated during bacterial infections were found to contribute to bacteria elimination but also to disease pathogenesis[26]. NK cell activation in these infections can occur both directly by sensing of bacteria through pattern recognition receptors and indirectly via bacterial stimulation of dendritic cells or macrophages[27]. In case of E. coli infection and its major pathogen-associated molecular pattern LPS, it was demonstrated that NK cell activation is facilitated via IL-2, IL-18 and IFN-ß produced by dendritic cells[28]. In view of the reported regulatory activity of NK cells, we hypothesized that bacterial coinfection may result in enhanced NK cell regulatory activity. Indeed, we here demonstrate that NK cells in LPS-treated mice suppress clonal expansion of LCMV-specific CTLs by a NKG2D-independent or NCR1-independent but perforin-dependent mechanism. These results suggest a TLR4-mediated immunoregulatory role of NK cells during viral-bacterial coinfections.

## Results

**E. coli coinfection interferes with LCMV control**. To determine whether a bacterial coinfection can interfere with LCMV-specific

CTL immunity, C57BL/6 (B6) mice were infected with a low dose (200 pfu) of LCMV (strain WE) followed by inoculation with $5 \times 10^5$ cfu of E. coli one day later. At day 8 post-infection (p.i.), the LCMV-specific CTL response was analyzed by MHC class I tetramer staining and by assessing viral titers. Without coinfection, the mice generated a robust virus-specific CTL response and decreased viral titer to low levels. Interestingly, coinfection with E. coli significantly reduced the LCMV-specific CTL response and strongly impaired virus elimination in spleen and liver. Most strikingly, antibody-mediated depletion of NK cells almost completely restored the LCMV-specific CTL response and virus clearance in E. coli coinfected mice (Fig. 1a–c).

To determine whether the negative effect of E. coli coinfection on anti-LCMV CD8 T cell immunity was mediated by bacterial cell wall components such as LPS or peptidoglycans, TLR2/4-deficient mice were used. In striking contrast to wild-type (wt) mice, E. coli coinfection ($2 \times 10^6$ cfu) of TLR2/4-deficient mice did neither inhibit the LCMV-specific CTL response nor impair viral clearance. In addition, depletion of NK cells before coinfection did not significantly improve the antiviral CTL response in these mice (Fig. 1d, e). Taken together, these data suggest that cell wall components released during E. coli infection enabled NK cells to interfere with induction of LCMV-specific CTL and virus clearance.

**TLR ligands inhibit the anti-LCMV CTL response**. To provide direct evidence that TLR4 triggering was able to interfere with the induction of LCMV-specific T cell immunity, purified LPS (1 μg) was injected into B6 mice that had been infected with LCMV one day earlier. At day 8 p.i., the LCMV-specific CTL response and viral titers were analyzed. Similar to the E. coli coinfection, LPS injection also strongly reduced the LCMV-specific CTL response and prevented rapid virus clearance. Importantly, NK cell depletion again reverted the negative effects of LPS treatment on induction of the LCMV-specific CTL response and virus elimination (Fig. 2a, c). Absolute numbers of splenocytes were comparable in all experimental groups indicating that gp33- and np396-tetramer+ CTL were de- or increased not only in relative but also in absolute numbers (Fig. 2b). Similar to LPS, injection of TLR3 ligand poly(I:C) or TLR9 ligand CpG oligodeoxynucleotides (ODN) also markedly suppressed the virus-specific T-cell response and virus clearance through a NK cell-dependent mechanism (Supplementary Fig. 1). LPS treatment did not suppress the anti-LCMV CTL response in TLR2/4-deficient mice. However, injection of poly(I:C) resulted in a decreased CTL response and impaired virus control, demonstrating that these mice were still responsive to NK cell activating signals (Supplementary Fig. 2). Taken together, these data show that TLR ligands when present at an early time point after LCMV infection strongly interfered with rapid viral clearance by a NK cell-dependent mechanism.

**NK cell-mediated inhibition of CTL is perforin-dependent**. To analyze the effect of LPS on the induction of LCMV-specific CTL at early time points, we used an adoptive transfer system with LCMV gp33-specific CD8 T-cells from P14 TCR transgenic mice. A tracer population of P14 T-cells (Thy1.1+) was transferred into B6 recipient mice followed by LCMV infection and LPS or PBS injection at day 1 p.i.. At day 4 after infection, P14 T-cell frequencies in spleens of infected mice were slightly increased compared to non-infected controls. This initial expansion was, however, not affected by the LPS treatment (Fig. 3a, left). Likewise, the rate of BrdU incorporation in P14 T cells was not influenced by LPS administration (Fig. 3a, right). These data indicate that LPS-treatment in the context of a LCMV infection

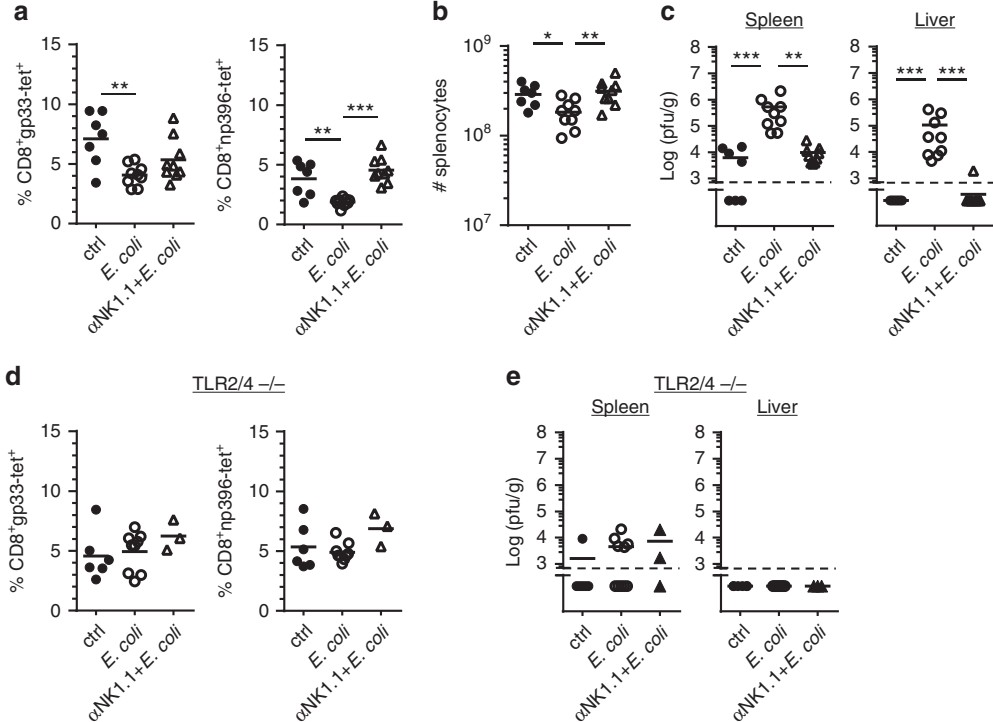

**Fig. 1** *E. coli* coinfection leads to NK cell-mediated impairment of the anti-LCMV CTL response. NK cell-depleted and non-depleted B6 (**a**–**c**) or TLR2/4-deficient mice (**d**, **e**) were infected with LCMV. One day later, they were coinfected with *E. coli* or received sterile LB medium as control (ctrl) and were analyzed at day 8 after LCMV infection. **a**, **d** gp33- and np396-tetramer$^{+}$ (tet) CD8 T cells in percent of spleen cells. **b** Absolute numbers (#) of splenocytes. **c**, **e** Viral titers in spleen and liver. Data of individual mice ($n = 7$–9 (**a**–**c**), $n = 3$–9 (**d**, **e**)) from 1–3 independent experiments with 2–4 mice per group are shown; horizontal bars represent the mean values. Dashed lines indicate detection limits. *$p < 0.05$, **$p < 0.01$, ***$p < 0.001$; one-way ANOVA with Tukey-Kramer post-test (**a**, **b**), Mann–Whitney (**d**, **e**), or Kruskal–Wallis Test with Dunn's post-test (**c**)

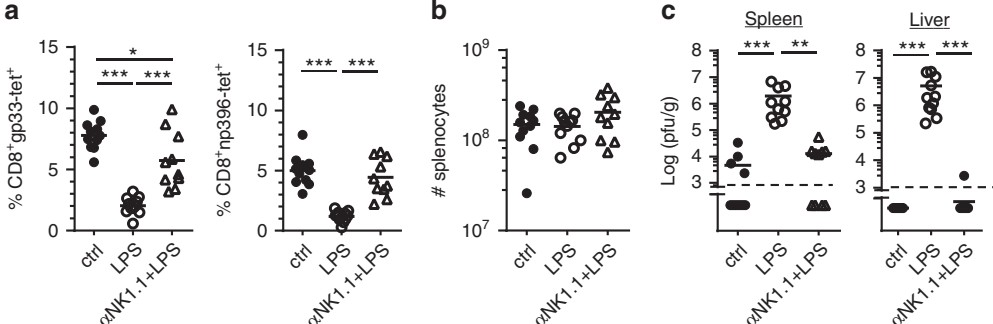

**Fig. 2** LPS treatment leads to NK cell-mediated impairment of the anti-LCMV CTL response. NK cell-depleted and non-depleted B6 mice were infected with LCMV. After one day of infection, they received LPS or PBS as a control (ctrl). Mice were analyzed at day 8 after LCMV infection. **a** gp33- and np396-tetramer$^{+}$ (tet) CD8 T cells in percent of spleen cells. **b** Absolute numbers (#) of splenocytes. **c** Viral titers in spleen and liver. Data of individual mice ($n = 11$) from 4 to 5 independent experiments with 2–3 mice per group are shown; horizontal bars represent the mean values. Dashed lines indicate detection limits. *$p < 0.05$, ** $p < 0.01$, ***$p < 0.001$; one-way ANOVA with Tukey-Kramer post-test (**a**, **b**) or Kruskal–Wallis Test with Dunn's post-test (**c**)

did not interfere with the initial priming of the CTL response. In sharp contrast to this, one day later (day 5 p.i.), frequencies and absolute numbers of clonally expanded P14 T-cells were considerably lower in LPS-treated mice compared to controls (Fig. 3b). As in the polyclonal setting, the LPS-induced decrease in P14 T-cell expansion was prevented by NK cell depletion (Fig. 3c). To confirm the importance of NK cells for the decreased expansion of P14 T cells after LCMV infection and LPS treatment, IL-15-deficient mice were used as recipients of P14 T cells. These mice almost completely lack NK cells but mount a fully functional CTL immune response after LCMV infection[29]. In contrast to wt recipient mice, LPS injection did not lower the

LCMV-induced expansion of P14 T cells in IL-15-deficient mice (Fig. 3d).

Regulation of T-cells by NK cells frequently operates through cell-mediated lysis via perforin[16,17,30]. To determine whether the decrease in expansion of P14 T-cells by LPS was perforin-dependent, perforin-deficient mice were used as recipients of P14 T cells. Unlike to wt recipients, LPS did not decrease the LCMV-induced expansion of P14 T-cells in hosts lacking perforin (Fig. 3e). For the LCMV clone 13 infection model, it was postulated that NK cells negatively regulate T cell priming by cell-mediated lysis of antigen presenting cells (APC)[19]. The unaltered initial P14 T cell expansion until day 4 p.i. (Fig. 3a), however,

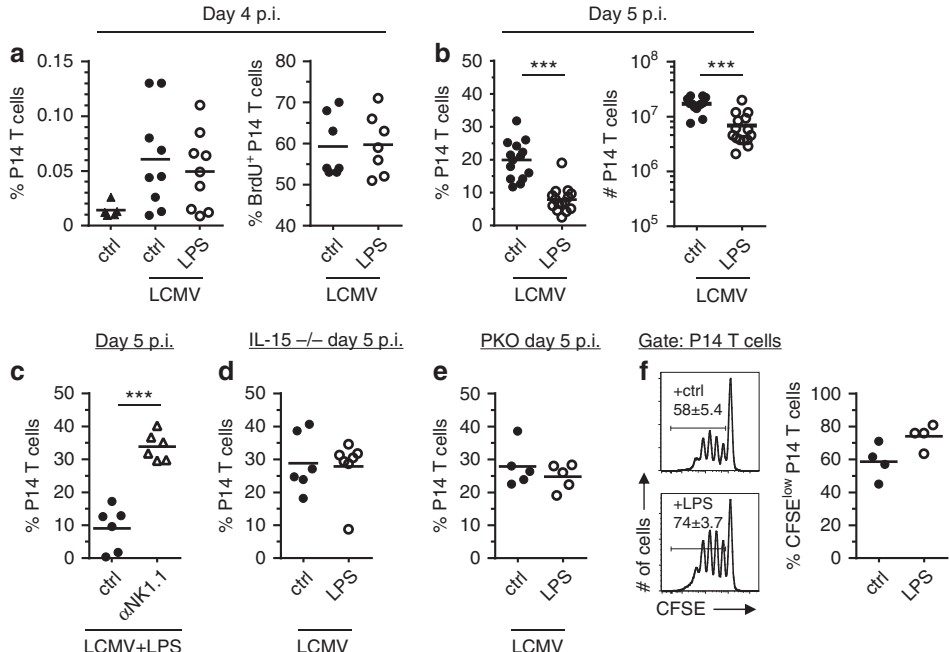

**Fig. 3** LPS-activated NK cells inhibit clonal expansion of LCMV-specific P14 CD8 T-cells. P14 T-cells were transferred into the indicated recipients which were subsequently infected with LCMV. After one day, they received LPS or PBS as a control (ctrl). **a** Percentages P14 T-cells of splenocytes (left) and BrdU+ cells of splenic P14 T cells (right) at day 4 p.i. or day 4.5 p.i. respectively. For BrdU labeling, mice were injected with BrdU and sacrificed 20 min later. Data are pooled from 2 to 4 independent experiments with 1–3 mice per group ($n = 5$–9). **b** Percentages P14 T cells of splenocytes (left) and absolute numbers (#) of P14 T-cells in spleen (right) at day 5 p.i.. Data are pooled from seven independent experiments with 1–3 mice per group ($n = 13$–15). **c** Percentages P14 T cells of splenocytes in NK cell-depleted and non-depleted B6 at day 5 p.i.. **d, e** Percentages P14 T cells of splenocytes in IL-15$^{-/-}$ and perforin$^{-/-}$ mice at day 5 p.i.. Data are pooled from 2 to 3 independent experiments with 2–3 mice per group ($n = 5$–7). **f** CFSE-labeled P14 T cells were stimulated with splenic APCs from LCMV-infected B6 mice (day 4 p.i.) that had been injected with LPS (1 μg) or PBS at day 1 p.i.. After 3 days, cell division of P14 T cells was analyzed by dye dilution. Representative histograms are shown, numbers in histograms indicate mean values ± SEM; dots represent mean values of duplicates. Data are pooled from two independent experiments ($n = 4$) with 1–3 mice per group. *** $p < 0.001$; Mann–Whitney Test

already suggested that APC capacity is unlikely to be affected in our setting. To further determine whether LPS injection interfered with priming of T cells, we used splenic APC from LCMV-infected mice (day 4 p.i.) to stimulate proliferation of naïve P14 T-cells. As depicted in Fig. 3f, APC isolated from LPS-treated mice showed an even slightly increased capacity to stimulate P14 T cell proliferation in vitro. Taken together, these data indicate that NK cells activated directly or indirectly by LPS during an ongoing LCMV infection negatively regulated proliferating LCMV-specific CD8+ T-cells by a perforin-dependent mechanism.

**LPS promotes NK cell accumulation after LCMV infection.** NK cells become activated during LCMV infection[31] but their numbers hardly increase (Fig. 4a). Interestingly, LPS given at day 1 after LCMV infection led to significantly increased NK cell frequencies and numbers in the spleen at day 4 p.i.. Likewise, NK cell numbers in liver and lungs were also considerably increased in LCMV/LPS- compared to LCMV/PBS-treated mice (Supplementary Fig. 3). Without infection, LPS treatment did not increase splenic NK cell frequencies and numbers. The cytolytic activity of LCMV/LPS-activated NK cells was tested in $^{51}$Cr release-assays using NK cell-sensitive YAC-1 target cells. Splenocytes from LCMV/LPS-treated mice showed a 3 to 5-fold increase in lytic activity against YAC-1 cells compared to splenocytes from LCMV-infected mice without LPS treatment (Fig. 4b, left). This difference was primarily due to the increased NK cell frequency since lytic activity of NK cells on a per cell basis was only slightly increased by LPS injection (Fig. 4b, right). NK cells from LPS-treated LCMV-infected mice also displayed a more mature phenotype with smaller CD11b$^-$CD27$^+$ but larger

CD11b$^+$CD27$^+$ subsets and increased KLRG1 expression (Fig. 4c).

To test whether direct TLR signaling in NK cells was required for LPS-induced cell proliferation during LCMV infection, CFSE-labeled NK cells from TLR2/4-deficient or wt mice were transferred into wt mice, followed by LCMV infection and LPS injection. After 4 days, cell division of the transferred NK cells was analyzed by CFSE dye dilution. LPS injection into LCMV-infected recipients significantly increased NK cell division as evident by an increased portion of CFSE$^{low}$ NK cells. This increase was also evident in NK cells lacking TLR2/4 (Fig. 4d). Thus, LPS given in the context of a LCMV infection promoted NK cell proliferation indirectly by a NK cell-extrinsic pathway.

**Role of IL-15 in negative T-cell regulation.** IL-15 is an important cytokine for NK cell survival and activation that is also induced by LPS[32,33]. To block IL-15 signaling, we used a monoclonal antibody (mAb) that specifically targets the IL-15/IL-15R complex. Without LPS treatment, blocking IL-15/IL-15R did not affect induction of LCMV-specific CTL and virus clearance (Fig. 5d–f) confirming previous findings that IL-15 is not required for effector T-cell induction[29]. Anti-IL-15/IL-15R mAb treatment, however, limited the increase of NK cells after LPS injection in the context of a LCMV infection (Fig. 5b). Nonetheless, anti-IL-15/IL-15R mAb treatment did not restore the LCMV-specific CTL response and virus elimination after LPS injection (Fig. 5a, c). Thus, IL-15 signaling was dispensable for the LPS-triggered negative T-cell regulation by NK cells. In addition, these data imply that the increased NK cell numbers observed after LPS injection were not a prerequisite for their suppressive effect on the LCMV-specific CTL response.

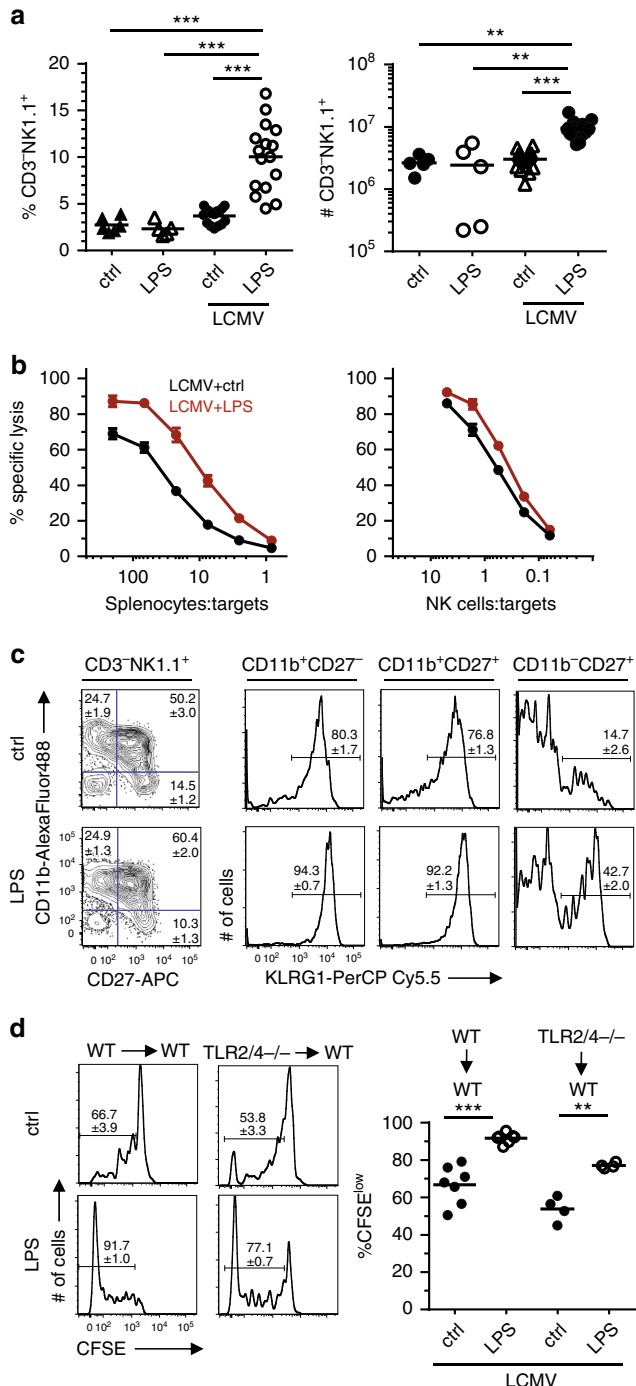

**Fig. 4** LPS promotes NK cell accumulation after LCMV infection. B6 mice were infected with LCMV, injected with LPS or PBS one day later and analyzed at day 4 p.i.. **a** Percentage and absolute numbers (#) of NK cells (CD3−NK1.1+) in spleens. Data are pooled from five independent experiments (n = 5–16) with 2–4 mice per group. **b** Specific lysis of YAC-1 target cells by NK cells. Results are displayed as total splenocyte-to target cell-ratio (left, n = 15) or as NK cell-to-target cell-ratio (right, n = 7). Data are pooled from 2 to 6 independent experiments with 2–4 mice per group. Dots represent mean values, error bars the SEM. **c** Expression of CD11b, CD27 and KLRG1, gated at the indicated NK cell subsets. Pooled data from five independent experiments (n = 11) with 2–4 mice per group and representative contour plots and histograms are shown. Numbers indicate means ± SEM. **d** CFSE-labeled wild-type (wt) or TLR2/4−/− NK cells were transferred into B6 recipient mice. One day later, recipient mice were infected with LCMV and another day later injected with LPS or PBS. At day 4 p.i., cell division of the transferred NK cells (CD45.2+CD3−NK1.1+) in spleen was analyzed by CFSE dye dilution. Representative histograms are shown, numbers indicate mean values ± SEM; dots represent values of individual mice. The positions of the gates were determined by the utmost right peak of the CFSE dilution histograms which represents undivided cells. The gates include all cells with lower fluorescence intensity when compared to the undivided CSFEhigh cells. Data are derived from five independent experiments with 1–3 mice per group (for wt NK cells, n = 7) and two independent experiments with two mice per group (for TLR2/4−/− NK cells, n = 4). *p < 0.05, **p < 0.01, ***p < 0.001; ANOVA with Tukey-Kramer post-test (**a**, left); Kruskal-Wallis with Dunn's post-Test (**a**, right); unpaired t-test with Welch-correction (**d**)

IFN-β, IL-2 and IL-18 have also been shown to be necessary and sufficient for NK cell activation following exposure to *E. coli* or LPS administration; induction of NK cell cytotoxic activity required IFN-β and IL-15 but not IL-2 or IL-18[28]. In our setting, however, IFN-β and IL-18 were dispensable for the suppressive effect of LPS-activated NK cells on the LCMV-specific T-cell response (Supplementary Fig. 4A to D). Also the combined deficiency of IL-18 and IL-12 did not prevent suppression of LCMV-specific T-cell immunity by LPS-activated NK cells (Supplementary Fig. 4E to F).

**NK cell-mediated inhibition is independent of NKG2D or NCR1.** NKG2D and NCR1 have both been reported to be responsible for killing of activated LCMV- and MCMV-specific CD8 T-cells by NK cells[16,23,30]. The importance of NKG2D in our model system was tested using NKG2D-deficient mice. The data show that LPS given in the context of a LCMV infection also strongly decreased the LCMV-specific CTL response and virus elimination in the absence of NKG2D (Fig. 6a–d). To assess the role of the activating NK cell receptor NCR1, we utilized two different mouse strains. First, similar to Crouse et al.[23], we used *NKp46*icre/icre mice which exhibit a strongly impaired NCR1 expression[34]. In these mice we found that LPS injection still suppressed the LCMV-specific CTL response and impaired virus elimination (Fig. 6e, g, h). However, NK cell frequencies in LCMV-infected *NKp46*icre/icre mice were considerably decreased when compared to wt mice (Fig. 6f). Therefore, we tested an additional recently described mouse line termed B6.CD45.1-*Ncr*C14R that lacks NCR1 cell surface expression due to a point mutation in the NCR1 signal peptide[35]. In these mice, NK cell frequencies were not affected by the lack of NCR1 expression (Fig. 6j). Nonetheless, LPS injection also significantly suppressed the anti-LCMV CTL response and increased viral burdens in the absence of NCR1 (Fig. 6i, j, l). Together, these results demonstrate that NKG2D and NCR1 were dispensable for LPS-induced NK cell-mediated suppression of the anti-LCMV CD8 T cell immune response.

## Discussion

Several studies have previously shown that NK cells are able to kill antigen-specific T-cells after LCMV infection. Importantly, this type of negative regulation of T-cells by NK cells in LCMV infection is only observed after inoculation with high virus doses (>10^4 pfu)[16,17,19]. When mice were infected with low doses (200 pfu) of LCMV-WE, as it was done in the present study, NK cell-depletion did not significantly improve the LCMV-specific CTL response and virus clearance[23–25]. Using this low-dose infection setting, we now demonstrate that TLR2/4 ligands generated during a coinfection with *E. coli* caused a NK cell-mediated

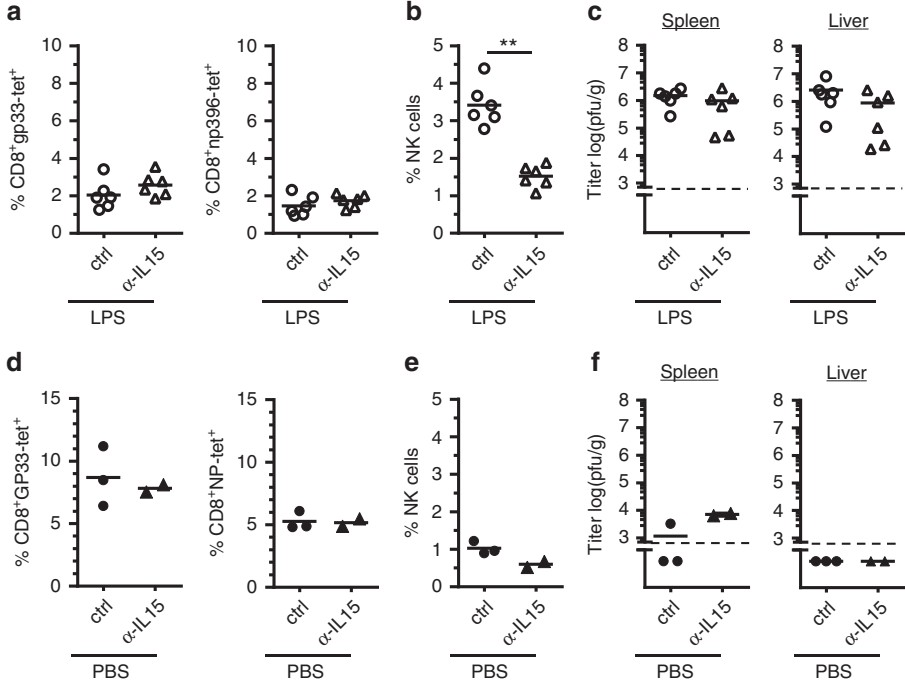

**Fig. 5** IL-15 blockade inhibits LPS-induced NK cell accumulation but not negative T-cell regulation. B6 mice were infected with LCMV and received LPS (**a–c**) or PBS (**d–f**) at day 1 p.i.. As indicated, groups of mice were additionally treated at day 1 and day 3 p.i. with mAb specific for IL-15/IL-15R complex. All mice were analyzed at day 8 after LCMV infection. **a, d** gp33- and np396-tetramer+ (tet) CD8+ T cells in percent of spleen cells in the indicated groups. **b, e** Percent NK cells (CD3−NK1.1+) of spleen cells. (**c, f**) Viral titers in spleen and liver. Data of individual mice from (**a–c**) two independent experiments with three mice per group ($n = 6$) and **d–f** one experiment with 2–3 mice per group ($n = 2–3$) are shown; horizontal bars represent the mean values. Dashed lines indicate detection limits. **$p < 0.01$, Mann–Whitney Test

suppression of the anti-viral CTL response and thereby prevented rapid virus control. Similarly, injection of the soluble TLR ligands LPS, poly(I:C) or CpG ODN also impaired the LCMV-specific CTL response and virus elimination through a NK cell-dependent mechanism. These findings reveal a hitherto unknown mechanism by which bacterial infections incapacitate a central antiviral effector pathway.

NK cells in LPS-treated mice suppressed the expansion of LCMV-specific P14 CD8 T-cells by a perforin-dependent mechanism. This suppression took place between day 4 and day 5 p.i., when the activated P14 cells underwent an extensive proliferative burst. Besides activated T-cells, LCMV antigen presenting cells (APC) may also serve as targets for NK cells[19]. However, LPS injection did not affect the capacity of ex vivo isolated splenic APCs from LCMV-infected mice to stimulate P14 T-cell proliferation in vitro. In addition, LPS treatment did not impair initial in vivo expansion of P14 T-cells up to day 4 p.i. and did not affect BrdU incorporation in P14 T-cells at day 4 p.i.. Hence, these data suggest that upon TLR ligand treatment NK cells kill proliferating virus-specific CD8 T-cells. In line with previous studies showing that CD4 T-cells can also serve as targets of NK cell regulatory activity[16,17], we observed that LPS-treatment decreased clonal expansion of LCMV-specific SMARTA CD4+ T-cells after LCMV infection as well (Supplementary Fig. 5). Since control of low dose LCMV infection is predominantly dependent on CD8+ T-cells, it is, however, unlikely that impaired virus elimination after LPS injection was primarily due to the missing LCMV-specific CD4+ T-cell response.

Dendritic cells are able to sense pathogens through pattern recognition receptors and, in turn, activate NK cells[36–38]. IL-15 induced by type I IFN-receptor signaling is further important for TLR4-triggered NK cell activation[28,37]. Antibody blockade of IL-15-signaling significantly impaired the LPS-induced

accumulation of NK cells during LCMV infection in our system. Nonetheless, this treatment failed to prevent their suppressive effect on the LCMV-specific CTL response. This indicates that the sole increase in NK cell numbers as observed in LCMV-infected mice after LPS injection was not a prerequisite for the suppression. Remarkably, IL-12, IL-18, and IFN-β were also dispensable. This suggests that the cytokine requirement for the inhibitory effect of NK cells on the LCMV-specific CTL response after TLR4 stimulation exhibits a considerable redundancy. This could be due to the two potent immunological stimuli, LCMV infection and LPS injection, used here.

Several receptors have been reported to be important for killing of activated CD4+ or CD8+ T-cells by NK cells. A crucial role of NKG2D was shown in two studies performed in the LCMV and in the MCMV model systems[16,30]. In contrast, Waggoner et al.[18] showed that absence of 2B4 promoted NK cell-mediated killing of LCMV-activated CD8+ T-cells but found no evidence for the involvement of NKG2D. Similar to Waggoner et al.[18] and Crouse et al.[23], we also did not observe a role of NKG2D in the negative regulation of LCMV-specific CD8+ T-cells by NK cells. The reason for this discrepancy is unknown and remains to be solved.

Absence of the type I IFN receptor (IFNAR) has been shown to render activated LCMV-specific CD8+ T-cells more susceptible to NK cell killing[23,25]. In addition, adoptive transfer experiments with IFNAR-deficient P14 T-cells into Ncr1icre/icre mice revealed an essential role of NCR1 in this process. In our experiments with Ncr1icre/icre mice, LPS treatment was still able to significantly decrease the LCMV-specific CD8+ T-cell response. However, the extent of this decrease as well as the increase in viral titers after LPS treatment was less pronounced in NKp46icre/icre mice than in wt mice. This could be due to the lower NK cell frequencies in NKp46icre/icre mice or non-defined strain differences. Importantly, LPS-treatment also significantly lowered the

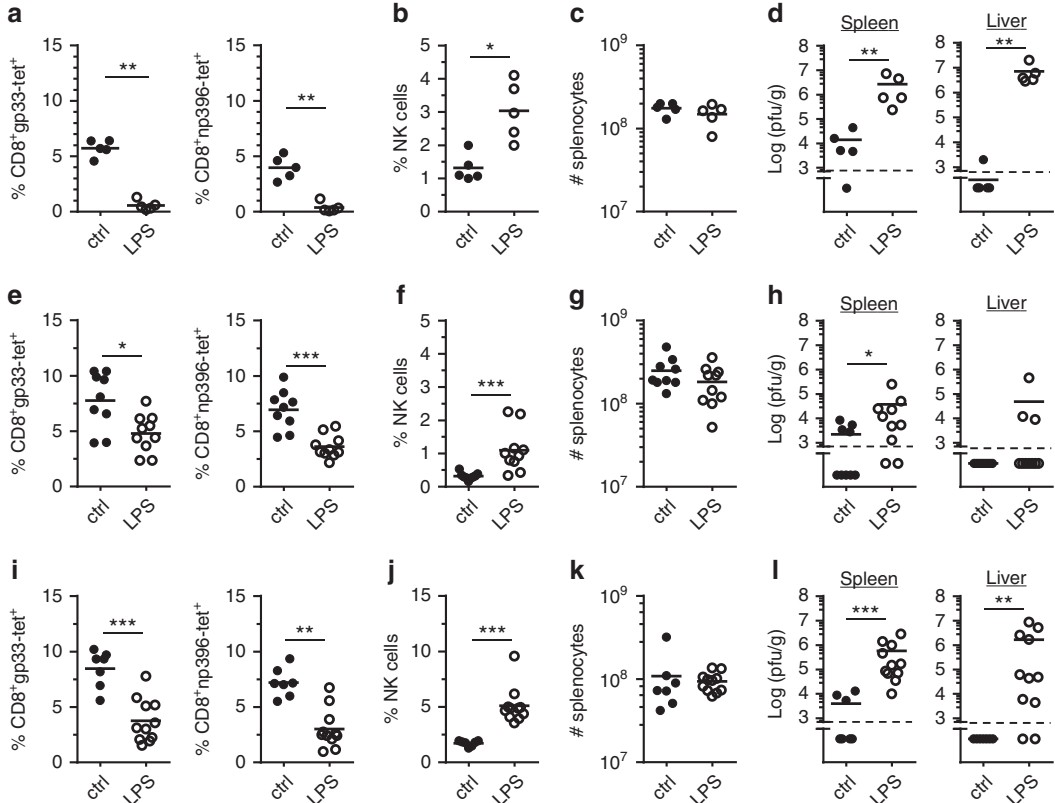

**Fig. 6** Deficiency in NCR1 or NKG2D does not prevent LPS-induced negative T cell regulation. NKG2D-deficient (**a–d**, n = 5), NKp46 icre/icre (**e–h**, n = 9–10) and B6.CD45.1-Ncr^C14R mice (**i–l**, n = 8–11) were infected with LCMV. One day later, they received LPS or PBS (ctrl). Mice were analyzed at day 8 after LCMV infection. **a**, **e**, **i** gp33- and np396-tetramer+ (tet) CD8 T cells in percent of spleen cells. **b**, **f**, **j** Percent NK cells (CD3−NK1.1+) of spleen cells. **c**, **g**, **k** Absolute numbers (#) of spleen cells. **d**, **h**, **l** Viral titers in spleen and liver. Data of individual mice from 2 to 3 independent experiments with 2–6 mice per group are shown; horizontal bars represent the mean values. Dashed lines indicate detection limits. *p < 0.05, **p < 0.01, ***p < 0.001; Mann–Whitney test

LCMV-specific CTL response and increased viral burdens in B6. CD45.1-Ncr^C14R mice that lack NCR1 cell surface expression but contain normal NK cell numbers[35]. Thus, these data indicate that NCR1 was not involved in the interaction of LPS-activated NK cells with activated LCMV-specific CD8+ T cells.

T-cells protect themselves from NK cell killing by expression of ligands for inhibitory NK cell receptors or by suppression of ligands for activating NK cell receptors[18,23,39]. For instance, type I IFNs are known to induce expression of MHC class I and Ib molecules that function as ligands for NK cell inhibitory receptors[25,40]. We observed that IFN-α and IFN-β levels in serum of LCMV-infected mice were roughly two-fold reduced at 24 h after LPS injection as compared to control mice. However, this decrease did not alter H-2K^b, H-2D^b and Qa-1b expression levels on LCMV-specific P14 CD8 T-cells isolated from NK cell-depleted hosts at day 5 p.i. (Supplementary Fig. 6). Thus, negative T-cell regulation by NK cells in LPS-treated mice occurred despite high expression of MHC class I and Ib molecules.

While the impact of viral infections on bacterial coinfections has been extensively studied in the past[41−44], the reverse regulation as described here has been poorly recognized to date. There are a few reports that previously demonstrated an effect of coinfection on LCMV-specific T-cell immunity. Polymicrobial sepsis analyzed in a cecal-ligation and puncture model has been shown to impair LCMV-specific T-cell immunity by various mechanism including loss of APC function, decrease in precursor T-cell frequencies and alterations in T-cell functions[9−12]. Mice co-infected with LCMV and *Schistosoma mansoni* further show enhanced early LCMV replication and impaired viral clearance in the liver most likely due to a decreased type I IFN response in this

organ[45]. Similarly, decreased type I IFN production that limits expansion of LCMV-specific CTL has been observed in mice co-infected with Ectromelia virus and LCMV[46]. Finally, coinfection of mice with Pichinde virus led to a decreased number of LCMV-specific CTL but increased variability in immunodominance that can be rationalized by the competition between two simultaneous immune responses[47].

In conclusion, the present study sheds light on an additional aspect in the complex network of viral-bacterial coinfections. In particular, it links TLR stimulation by bacterial pathogens to NK cell-mediated suppression of virus-specific T cells, which has not been reported previously. The results have important implications for our understanding of the immune defense and mechanisms of evasion when the organism is simultaneously exposed to two different pathogens.

## Methods

**Mice**. C57BL/6 (B6) mice were obtained from Janvier (Le Genest St-Isle, France). B6-Ly5.2/Cr (CD45.1+), IL-15-deficient[48], IL-18-deficient[49], perforin-deficient[50], TLR2/4-deficient[51], NKG2D-deficient[52], NKp46^icre/icre[53], IFN-β^luc/luc[54], P14 TCR tg[55], SMARTA TCR tg[56] and B6.CD45.1-Ncr^C14R[35] mice were bred locally. P14 chimeric mice were generated by adoptive transfer (i.v.) of 1 × 10^5 splenic P14 TCR+ tg T-cells into B6 mice. Mice were bred and kept in our animal facility under specific pathogen-free conditions. Animal care and use was approved by the Regierungspräsidium Freiburg. All experiments were performed in accordance with the German law for animal protection.

**Infections and treatments**. Mice were infected i.v. with 200 pfu LCMV-WE. For coinfection experiments, B6 and TLR2/4-deficient mice were infected one day later with 5 × 10^5 or 2 × 10^6 cfu E. coli EH100, respectively, in Lysogeny Broth (LB) medium (200 μl, i.v.). E. coli were grown in LB medium and frozen at −80 °C in glycerol stocks. Before infection, E. coli were regrown in LB medium until an

optical densitiy (OD 600 nm) of 0.4 was reached. Infection dose of bacteria was calculated from the linear relationship between turbidity and the number of cfu grown after 24–48 h at 37 °C. Control mice received sterile LB medium. For TLR-ligand treatment, mice received one day after LCMV infection 1 µg LPS derived from *E. coli* O111:B4 (Sigma-Aldrich), 50 µg poly(I:C) (Enzo) or 50 µg CpG ODN 1668 (InvivoGen) in 200 µl PBS via the intravenous route; controls received PBS only. NK cells were depleted by injection of 300 µg and 200 µg purified anti-NK1.1 mAb (clone PK136, BioXcell) at day 2 and day 1, respectively, before LCMV infection. In the P14 T-cell transfer experiments (Fig. 3c), control (ctrl) mice received mouse IgG2a isotype control mAb (clone C1.18.4, BioXcell). Since treatment with isotype control mAb had no effect on the anti-LCMV CTL response and virus clearance (Supplementary Fig. 7), isotype control antibodies were omitted for the NK cell-depletion experiments without P14 T-cell transfers (Fig. 1 and Fig. 2). For blocking IL-15 signaling, mice were treated at day 1 and 3 after infection (i.p.) with anti-IL-15/IL-15R α chain complex antibody (50 µg, clone GRW15PLZ, eBioscience) or isotype control mAb (clone HRPN, BioXcell) or left untreated. For blocking IL-12, anti-IL-12 p40 antibody (200 µg, clone C17.8, BioXcell) was given (i.p.) at day 0 and at day 1 and 3 after infection.

**Flow cytometry**. For isolation of lymphocytes, spleens were meshed through a metal strainer. Livers were meshed through a cell strainer (Greiner) and lymphocytes purified using a Percoll (Sigma) gradient. Lungs were digested using Collagenase II (Roche, 140 U ml$^{-1}$) and DNaseI (Sigma,10 µg ml$^{-1}$) and meshed through a cell strainer. The following mAbs were purchased from BioLegend (BL), eBioscience (eB) or Miltenyi Biotec (MB): anti-CD8α (clone 53.6.7, cat. # 100723 (BL), diluted 1:100), anti-Thy1.1 (HIS51, #14-0900-81 (eB), 1:1000), anti-CD3ε (145-2C11, #100312, 1:100), anti-CD11b (M1/70, #101217 (BL), 1:100), anti-CD27 (LG.7F9, #124211 (BL), 1:500), anti-KLRG1 (2F1, #138418 (BL), 1:150), anti-CD45.1 (A20, #110724 (BL), 1:200), anti-NK1.1 (PK136, #108714, 1:500), anti-CD4 (GK1.5, #47-0041-82 (eB), 1:100), anti-H-2K$^b$ (AF6-885, #116505 (BL), 1:100), anti-H-2D$^b$ (KH95, #111508 (BL), 1:50), anti-Qa-1B (6A8.6F10.1A6, #13$^-$105-048 (MB), 1:10). Zombie NIR dye (BioLegend) or DAPI (Sigma) was used for dead cell exclusion. To detect virus-specific CD8$^+$ T cells, lymphocytes were stained with D$^b$GP33 and D$^b$NP396 tetramers[57] (produced in-house). Staining was performed for at least 20 min at 4 °C. Samples were measured on a LSR Fortessa or Canto II cytometer (both BD Biosciences) and data were analyzed with FlowJo software 8.8.7 (Tree Star).

**$^{51}$Chromium-release assay**. Cytolytic activity of NK cells was determined by a standard $^{51}$Chromium-release assay. In brief, serial 1:3-dilutions of effector cells were mixed in 96-well round bottom plates with YAC-1 target cells (obtained from Dr. Rolf Zinkernagel, Zürich) that had been loaded with $^{51}$Chromium (Perkin-Elmer) for 2 h at 37 °C. Total splenocytes or enriched NK cells (MojoSort NK cell isolation kit, BioLegend) were used as effector cells. After incubation at 37 °C for 5 h, radioactivity in the supernatant was measured using a gamma-counter. Duplicate wells were assayed for each effector-target ratio and percentages of specific lysis were calculated.

**In vitro T-cell stimulation assay**. To test antigen presenting cell (APC) function, spleens from LPS-treated and control-treated LCMV-infected mice at day 4 p.i. were digested with collagenase II (Roche, 2000 U/ml) for 30 min at 37 °C and subsequently minced through a metal strainer. Afterwards, $2 \times 10^5$ splenocytes depleted of Thy1.2$^+$ and B220$^+$ cells by positive selection (Dynabeads Magnetic Separation, ThermoFisher) were co-cultured with $2 \times 10^5$ CFSE-labeled enriched (mouse CD8 T-cell isolation kit, Miltenyi Biotec) P14 TCR$^+$ T cells in in 96-well plates for 3 days. Afterwards, cell division of P14 T-cells was analyzed by dye dilution.

**NK cell proliferation in vivo**. NK cells were enriched from CFSE-labeled splenocytes of indicated donor mice using mouse CD49b-Microbeads (Miltenyi Biotec) or the MagniSort Mouse NK cell isolation kit (eBioscience). $3 \times 10^5$ to $8 \times 10^5$ enriched CFSE-labeled NK cells were transferred one day prior to infection into indicated mice expressing a different CD45 isoform.

**Viral titers**. Viral titers were determined by standard focus-forming assay[58]. In brief, organs were homogenized using a FastPrep-24 (MPBiomedicals). Serial 1:10-dilutions of tissue homogenate were plated on MC57G fibrosarcoma cells (obtained from Dr. Rolf Zinkernagel, Zürich) in 24-well plates and after 4 h incubation at 37 °C, an overlay containing 1% methylcellulose was added. After another incubation for 40 h at 37 °C, supernatant was discarded and cells were fixed with 4% formaldehyde in PBS, followed by permeabilization using 0.5 % Triton X-100 in PBS, blocking with 10% FCS in PBS and staining with anti-LCMV NP mAb (clone VL-4, made in-house) and horse radish peroxidase-conjugated polyclonal goat-anti-rat IgG antibody (Jackson ImmunoResearch) as secondary antibody. Foci were detected by incubation with SIGMAFAST OPD (Merck).

**Serum type I IFN concentrations**. Mouse sera were obtained using BD Micro-tainer SST tubes. IFN-α concentration in the sera was determined using the

VeriKine Mouse Interferon Alpha ELISA Kit (pbl), IFN-β concentration was determined using the BD Cytometric Bead Array Mouse inflammation Kit, both according to the manufacturers instructions.

**Statistics**. Statistical differences between two groups were determined using unpaired two-tailed *t*-test or Mann–Whitney test depending on whether data demonstrated Gaussian distribution or not. When data were sampled from Gaussian distribution but had different standard deviations, unpaired two-tailed *t*-test with Welsh correction was used. Statistical differences between more than two groups were determined using two-tailed one-way ANOVA with Tukey-Kramer post-test or two-tailed Kruskal–Wallis test with Dunn's post-test depending on whether data demonstrated Gaussian distribution or not. All tests were performed using the GraphPad InStat software.

## Data availability

All relevant data are available from the authors upon request.

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

## Acknowledgements

The work was supported by the Deutsche Forschungsgemeinschaft DFG (SFB 1160/P3 to H.P. and SP1937 to G.G.). We thank Prof. Eric Vivier for providing NKp46^icre/icre mice and Dr. Peter Aichele for critical comments on the manuscript.

## Author contributions

T.S. designed, planed and performed the experiments, analyzed data and wrote the paper. M.A.F., U.S., C.B. and G.G. provided reagents or mice and were involved in data discussion and drafting the manuscript. H.P. initiated, organized and designed the study, wrote the paper and completed the manuscript.

## Additional information

**Competing interests:** The authors declare no competing interests.

