## [Peer Review File · Nature Communications]

Reviewers' comments:

Reviewer #1 (Remarks to the Author):

This is an outstanding paper that provides significant new insights into the phenomenon of NK cell regulation of immune responses during virus infection. The authors use a refined and highly relevant model that couples a low dose of virus with bacterial stimuli to provide a new, translatable paradigm about when and how NK cell suppression of T cells occurs. The authors also provide some pertinent mechanistic insights into the receptors that drive this activity. My concerns are mostly minor.

1. As presented, the T cell measurements in TLR-KO mice in Figure 1D/E already look low compared to wild-type mice, even before addition of *E. coli*. It would be prudent to stimulate the TLR-KO mice with polyI:C (Supplemental Figure 1) to show they are still responsive to an NK cell-inducing signal, and to deplete the *E. coli*-treated TLR-KO mice of NK cells to show that T cell responses are not elevated, as additional support for the conclusions drawn.

2. There is no indication that a control antibody was used in non-depleted mice as a control for the 500 micrograms of anti-NK1.1 that is injected into depleted mice. Either this is an oversight in detailing the methods, or there is a need to repeat one or more key NK cell depletion experiments with inclusion of a proper isotype control treatment group.

3. It would be worthwhile to perform a small experiment adding TLR-KO mice to Fig 2 to ensure that the effects of LPS are abrogated as well (especially given results in Figure 4D).

4. The impressive LPS-induced expansion of NK cells is intriguing, given that it is independent of T cell suppression. Additional data on NK cell frequencies in liver or other viral target tissues would be valuable to evaluate whether altered homing also contributes to increase in splenic NK. How is position of gate in Figure 4D determined?

5. Supplemental Figure 2 should be incorporated into Figure 5 since it is important to proper interpretation of that data.

6. There is no statistics section in the Methods. A variety of statistical tests were used and I would like to see justification of these disparate choices.

7. Ref 36 on line 303 looks the same as Ref 16.

Refereed by Stephen Waggoner
Assistant Professor of Pediatrics
Cincinnati Children's Hospital

Reviewer #2 (Remarks to the Author):

In this study, the authors have examined the impact of gram-negative bacterial infection (*E. coli*) or bacterial LPS on the outcome of LCMV infection. They demonstrate that co-infection or administration of LPS reduce LCMV-specific T cell responses and increase viral load. They demonstrate that reduced T cell responses after co-infection are not observed in TLR2/4-deficient mice and that NK cells are responsible for LPS-induced T cell suppression.

Comments

In Figure 4 B, the authors used an E:T ratio that is based on splenocytes:targets but they should have the frequency of peptide-specific T cells and it would be better to graph this data based on

the true Effectors (i.e., peptide-specific CD8⁺ T cells-to-target ratios) rather than being based on total splenocyte numbers.

Point-to-point-reply

Reviewer #1

This is an outstanding paper that provides significant new insights into the phenomenon of NK cell regulation of immune responses during virus infection. The authors use a refined and highly relevant model that couples a low dose of virus with bacterial stimuli to provide a new, translatable paradigm about when and how NK cell suppression of T cells occurs. The authors also provide some pertinent mechanistic insights into the receptors that drive this activity. My concerns are mostly minor.

1. As presented, the T cell measurements in TLR-KO mice in Figure 1D/E already look low compared to wild-type mice, even before addition of *E. coli*. It would be prudent to stimulate the TLR-KO mice with poly(I:C) (Supplemental Figure 1) to show they are still responsive to an NK cell-inducing signal, and to deplete the *E. coli*-treated TLR-KO mice of NK cells to show that T cell responses are not elevated, as additional support for the conclusions drawn.

As suggested, TLR2/4-deficient mice were also treated with poly(I:C) after LCMV infection. The results (new Supplementary Fig. 2) show that in contrast to LPS, poly(I:C) was able to decrease the LCMV-specific CTL response and to impair virus control in these mice. As further proposed, NK cells in LCMV-E.coli coinfecting TLR2/4-deficient mice were also depleted. As expected, this NK cell depletion did not result in an improved anti-LCMV immune response (Figure 1 D and E). In sum, the results of these control experiments further confirmed our previous conclusions.

2. There is no indication that a control antibody was used in non-depleted mice as a control for the 500 micrograms of anti-NK1.1 that is injected into depleted mice. Either this is an oversight in detailing the methods, or there is a need to repeat one of more key NK cell depletion experiments with inclusion of a proper isotype control treatment group.

In the P14 T cell transfer experiments shown in Figure 3 C, the control (ctrl) group received the same amount of isotype control antibodies. In the NK cell-depletion experiments depicted in Figure 1 and 2, we did not use isotype control antibodies. As recommended by the reviewer, we repeated a key experiment using isotype control monoclonal antibodies (mAb) for the non-depleted LPS-treated group. The results of this experiment (shown below) are almost superimposable to those depicted in Figure 2. This indicated that antibody treatment per se had no impact on the antiviral immune response. This information has been added to the Material and Methods section.

B6 mice, treated with anti-NK1.1 mAb or isotype control mAb, were infected with LCMV. After one day of infection, they received LPS or PBS as a control (ctrl). Mice were analyzed at day 8 after LCMV infection. (A) gp33- and np396-tetramer⁺ (tet) CD8 T cells in percent of spleen cells. (B) Viral titers in spleen and liver.

3. It would be worthwhile to perform a small experiment adding TLR-KO mice to Fig 2 to ensure that the effects of LPS are abrogated as well (especially given results in Figure 4D). *The experiment was performed as suggested. The results are shown in Supplementary Fig. 2. Similar to E.coli coinfection, LPS-treatment had no impact on the LCMV-specific CTL response and virus clearance in TLR2/4-deficient mice.*

4. The impressive LPS-induced expansion of NK cells is intriguing, given that it is independent of T cell suppression. Additional data on NK cell frequencies in liver or other viral target tissues would be valuable to evaluate whether altered homing also contributes to increase in splenic NK. How is position of gate in Figure 4D determined?

As suggested, we also determined the numbers of NK cells in liver and lungs at day 4 after LCMV infection in LPS-treated and control mice. As for the spleen, the numbers of NK cells in liver and lungs of LCMV/LPS-injected mice were increased when compared to LCMV-infected mice. These new data are shown in Supplementary Fig. 3. The positions of the gates in Figure 4 D were determined by the utmost right peak of the CFSE dilution histogram which represents undivided cells. The gates include all cells with lower fluorescence intensity when compared to the undivided CFSE^{high} cells. This information is now included in the corresponding figure legend.

5. Supplemental Figure 2 should be incorporated into Figure 5 since it is important to proper interpretation of that data.

We agree and incorporated former Supplementary Fig. 2 into Figure 5.

6. There is no statistics section in the Methods. A variety of statistical tests were used and I would like to see justification of these disparate choices.

A "statistics" part was added to Material and Methods section including justifications with regard to the use of the particular tests.

7. Ref 36 on line 303 looks the same as Ref 16.

The mistake was corrected.

Reviewer #2

In this study, the authors have examined the impact of gram-negative bacterial infection (E. coli) or bacterial LPS on the outcome of LCMV infection. They demonstrate that co-infection or administration of LPS reduce LCMV-specific T cell responses and increase viral load. They demonstrate that reduced T cell responses after co-infection are not observed in TLR2/4-deficient mice and that NK cells are responsible for LPS-induced T cell suppression.

Comments

In Figure 4 B, the authors used an E:T ratio that is based on splenocytes:targets but they should have the frequency of peptide-specific T cells and it would be better to graph this data based on the true Effectors (i.e., peptide-specific CD8+ T cells-to-target ratios) rather than being based on total splenocyte numbers.

We agree with reviewer that it is important to graph the data based on true effectors.

Actually, we already presented the data in this manner in the first version: Figure 4 B (left) depicted the total splenocyte-to-target ratio whereas Figure 4 B (right) showed the NK cell-to-

target ratio. Nonetheless, we now made this more clear the the figure legend. We also would like to point out that the lytic activity of NK cells and not CD8⁺ T cells was tested in Figure 4B.

REVIEWERS' COMMENTS:

Reviewer #1 (Remarks to the Author):

The authors were diligent in responding to inquiries and concerns with new data that bolster the conclusions. The paper is improved and remains an outstanding source of significant new insights into the phenomenon of NK cell regulation of immune responses during virus infection. I have no remaining concerns.

Reviewer #2 (Remarks to the Author):

The authors have addressed my previous concerns

REVIEWERS' COMMENTS:

Reviewer #1 (Remarks to the Author):

The authors were diligent in responding to inquiries and concerns with new data that bolster the conclusions. The paper is improved and remains an outstanding source of significant new insights into the phenomenon of NK cell regulation of immune responses during virus infection. I have no remaining concerns.

Reviewer #2 (Remarks to the Author):

The authors have addressed my previous concerns

Reply: We wish to thank both reviewers for their time and their constructive comments that have been a great help to improve our manuscript.